# UNDERSTANDING LLAVA'S VISUAL QUESTION AN-SWERING IN A MECHANISTIC VIEW

## ABSTRACT

Understanding the mechanisms behind Large Language Models (LLMs) is crucial for designing improved models and strategies. While recent studies have yielded valuable insights into the mechanisms of textual LLMs, the mechanisms of Multi-modal Large Language Models (MLLMs) remain underexplored. In this paper, we apply mechanistic interpretability methods to analyze the visual question answering (VQA) mechanisms in the first MLLM, Llava. We compare the mechanisms between VQA and textual QA (TQA) in color answering tasks and find that: a) VQA exhibits a mechanism similar to the in-context learning mechanism observed in TQA; b) the visual features exhibit significant interpretability when projecting the visual embeddings into the embedding space; and c) Llava enhances the existing capabilities of the corresponding textual LLM Vicuna during visual instruction tuning. Based on these findings, we develop an interpretability tool to help users and researchers identify important visual locations for final predictions, aiding in the understanding of visual hallucination. Our method demonstrates faster and more effective results compared to existing interpretability approaches. Our code, data and interpretability tool will be made available on GitHub.

## 1 INTRODUCTION

Large Language Models (LLMs) (Brown, 2020; Ouyang et al., 2022; Chowdhery et al., 2023; Touvron et al., 2023) have achieved remarkable results in numerous downstream tasks, including reading comprehension (Xiao et al., 2023), question answering (Tan et al., 2023), and sentiment analysis (Deng et al., 2023). However, their interpretability remains limited, and the underlying mechanisms are not yet well understood. This lack of clarity poses a significant challenge for researchers attempting to address issues such as hallucination (Yao et al., 2023), toxicity (Gehman et al., 2020), and bias (Kotek et al., 2023) in LLMs. Therefore, understanding the mechanisms of LLMs has become an increasingly important area of research. Recently, efforts have been made to explore the mechanisms behind different LLM capabilities, including factual knowledge (Meng et al., 2022; Geva et al., 2023), in-context learning (Wang et al., 2023; Wei et al., 2023), arithmetic (Stolfo et al., 2023), and reasoning (Wang & Zhou, 2024).

Although numerous studies have explored the mechanisms of LLMs, they have predominantly focused on textual LLMs, often overlooking Multi-modal Large Language Models (MLLMs). It has been demonstrated that features from different modalities, such as images and audio, can significantly enhance the core abilities of LLMs (Zhang et al., 2024). Therefore, investigating the mechanisms of MLLMs is essential. In this paper, we examine the mechanism of visual question answering (VQA) in Llava (Liu et al., 2024b), marking the first attempt to extend visual instruction tuning within an existing textual LLM, Vicuna (Chiang et al., 2023). As illustrated in Figure 1(a), Llava takes an image $X_v$ and a question $X_q$ as input. The image patches (a series of sub-figures of the input image) are transformed into a sequence of image embeddings $H_v$ by a projection matrix $W$ within a visual encoder, CLIP (Radford et al., 2021). Simultaneously, the question is transformed into a series of word embeddings by the embedding layer. The transformed image embeddings and question word embeddings are then processed by the model to generate the final answer $X_a$. Our study seeks to address three key questions: a) What is the relationship between the mechanisms of VQA and textual QA (TQA)? b) Are the visual features interpretable under textual LLM's interpretability analysis method? c) How does Llava acquire its VQA ability during visual instruction tuning?

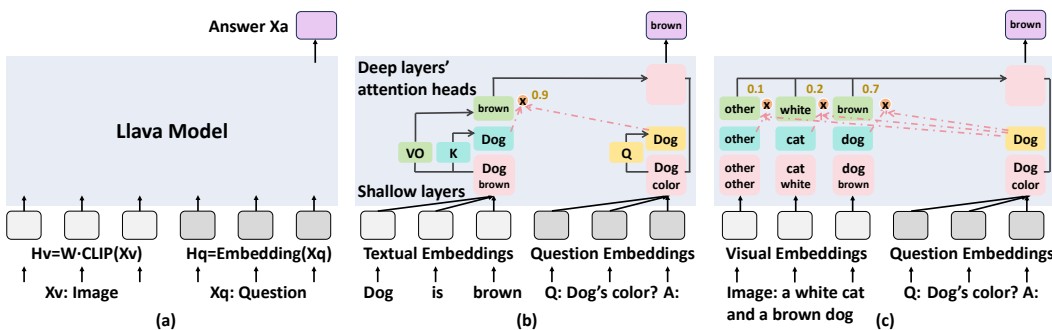

Figure 1: (a) Overall structure of Llava for VQA. The input of Llava is an image and a question. The image $X_v$ is transformed into image embeddings $H_v$ by a projection $W$ and the CLIP visual encoder. The question $X_q$ is transformed into question embeddings $H_q$ by the embedding layer. The model generates the answer $X_a$ based on $H_v$ and $H_q$. (b) Mechanism of textual QA in Vicuna. In shallow layers, the color position ('brown') extracts the animal features ('dog'). In deep layers' attention heads, the value-output matrices extract the color features ('brown') and the query-key matrices compute the similarity score between the last position (encoding the question about dog) and the color position's features ('dog'). The larger the similarity score, the higher probability of the final prediction 'brown'. (c) Mechanism of visual QA. The visual embeddings already contain the color features (brown, white) and the animal features (dog, cat). In deep layers' attention heads, the value-output matrices extract the color features and the query-key matrices compute the similarity between the last position (encoding the question about dog) and each position (encoding dog/cat).

We explore these questions on the color answering task, as color information is one of the most prominent features in images, making it an ideal starting point for understanding the mechanism of VQA. For VQA, we collect animal photos from the COCO dataset (Lin et al., 2014) featuring an animal [A] and its correct color [C], and then ask 'What is the color of [A]?'. For TQA, we generate a corresponding textual context for each photo, such as '[A] is [C].' For instance, the complete input might be 'Dog is brown. Q: What is the color of the dog? A:', with the correct answer 'brown'. We investigate the TQA mechanism in Vicuna, as shown in Figure 1(b). In shallow layers, the color position ('brown') extracts the animal information ('dog'). In deep layers' attention heads, the value-output matrices extract the color features ('brown'), while the query-key matrices compute the similarity between the last position (encoding the question about 'dog') and the color position's animal features, determining how much the probability of 'brown' is increased. When the question involves the same animal as the textual context, the attention score at the color position is large, and this position shows the largest log probability increase among all positions because it contains substantial information relevant to the final prediction 'brown'.

Next, we investigate the mechanism of VQA in Llava, starting by using log probability increase scores to identify the most important image regions, which we find to be the image patches related to the animals (as shown in Figure 2). We then apply the same methods used in textual question answering (TQA) to analyze the value-output matrices and the query-key matrices for these key output vectors. Our analysis reveals that the VQA mechanism in Llava is similar to that of TQA: the value-output matrices extract color information, while the query-key matrices compute the similarity between the question content and the animal features. Further, we employ interpretability methods to analyze the visual features in the embedding layer by projecting them into the embedding space (Dar et al., 2022). We discover that the visual embeddings exhibit significant interpretability regarding colors and animals, indicating that these embeddings already contain essential information about both. Based on these findings, we propose the hypothesis for the VQA mechanism shown in Figure 1(c). This hypothesis suggests that the visual embeddings store information about animals and colors, which is then transferred to deeper layers via the positions' residual streams. In the deep layers' attention heads, the value-output matrices extract color features, while the query-key matrices calculate the similarity between the question and the animal features. Finally, we compare the most important heads across vicuna TQA, Llava TQA, and Llava VQA, finding that the important attention heads are similar in all scenarios. This result suggests that Llava enhances Vicuna's existing abilities during visual instruction tuning.

According to these findings, we propose an interpretability tool for users and researchers to understand the important image patches that influence final predictions in Llava's VQA (Figure 6), which is helpful for understanding visual hallucination. Existing studies typically rely on causal explanations (Rohekar et al., 2024) or average attention scores (Stan et al., 2024) to locate important visual features. However, causal explanation methods require much computational cost, and average attention scores lack strong interpretability. Comparatively, our method computes the log probability increase at each position to identify the important locations in visual features, achieving much lower computational cost than causal explanations and much better interpretability than average attention.

| input image | log probability increase | avg attention score | log probability increase | avg attention score |
| :---: | :---: | :---: | :---: | :---: |
| 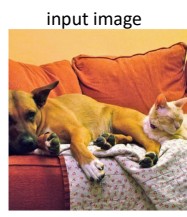 | 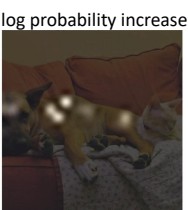 | 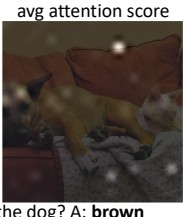 | 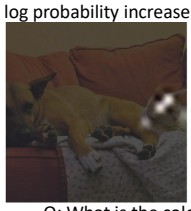 | 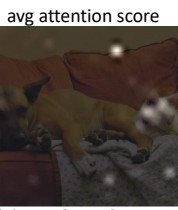 |
| | Q: What is the color of the dog? A: **brown** | | Q: What is the color of the cat? A: **white** | |

Figure 2: Identifying important image patches related to final predictions.

Overall, our contributions are as follows:

1) We explore the mechanism of Textual QA in Vicuna and Visual QA in Llava. We find that the visual embeddings are interpretable when projecting into embedding space. We find Llava enhances Vicuna's existing abilities during visual instruction tuning.

2) Based on the analysis of the mechanism of VQA, we design an interpretability tool to understand the important locations for final predictions, helpful for understanding visual hallucination. Compared with previous methods, our method achieves better interpretability and lower computational cost, which can be used for real-time interpretations.

## 2 RELATED WORKS

### 2.1 UNDERSTANDING TEXTUAL LLMS

Causal intervention (Vig et al., 2020) is a common method for identifying important modules in LLMs (Zhang & Nanda, 2023; Makelov et al., 2023), by computing the change of the final prediction when intervening the module. Using causal intervention, Meng et al. (2022) find the medium MLP layers in GPT2 store important parameters for knowledge. Stolfo et al. (2023) find similar stages in arithmetic tasks. A serious of studies (Merullo et al., 2023; Lieberum et al., 2023) focus on constructing the internal circuit in transformers from input to output, taking the attention heads and MLP layers as basic units. Elhage et al. (2021) and Olsson et al. (2022) find that the induction heads are helpful for predictions like [A][B] ... [A] => [B]. Hanna et al. (2024) explore how GPT2 computes greater-than algorithm. Gould et al. (2023) find the successor heads help predict the next number like Monday => Tuesday. Wang et al. (2022) study how GPT2 performs the indirect object identification task. Prakash et al. (2024) investigate the circuit between and after fine-tuning and find that fine-tuning enhances existing mechanisms. Conmy et al. (2023) propose a method to construct the circuits automatically. Another type of works aim to explore the neurons' interpretability (Dai et al., 2021; Sajjad et al., 2022; Nanda et al.; Gurnee et al., 2023). Geva et al. (2022) find that the MLP neurons are interpretable when projecting into the unembedding space. Dar et al. (2022) observe that other vectors are also interpretable in the unembedding space. Yu & Ananiadou (2024) calculate log probability increase to identify the important modules for the predictions.

### 2.2 UNDERSTANDING MULTIMODAL LLMS

Compared with textual LLMs, only a few studies have investigated the mechanisms of MLLMs. Stan et al. (2024) design a interpretability tool for vision-language models using average attention, relevancy map and causal interpretation. Basu et al. (2024) apply causal intervention methods to understand the information storage and transfer in MLLMs. Tong et al. (2024) study the shortcomings of the visual encoder CLIP. Gandelsman et al. (2023) explore the interpretability of CLIP.

## 3 Mechanism Explorations of Textual QA and Visual QA

In this section, we investigate the mechanism of VQA. We provide the background in Section 3.1, followed by an exploration of the mechanisms of TQA (Section 3.2) and VQA (Section 3.3). Finally, we compare the important attention heads before and after visual instruction tuning in Section 3.4.

### 3.1 Background

**Inference pass of decoder-only LLMs.** Except the visual encoder and the projection matrix, Llava and Vicuna has the same decoder-only LLM architecture as Llava is a fine-tuned model of Vicuna. Therefore, we start from introducing the inference pass of decoder-only LLM with textual inputs. Given $X = [x_1, x_2, ..., x_T]$ with $T$ tokens, the model predicts an output distribution $Y$ over $B$ tokens in vocabulary $V$. Every token $x_i$ (at position $i$) is transformed into a word embedding $h_0^i \in \mathbb{R}^d$ by embedding matrix $E \in \mathbb{R}^{B \times d}$. After that, the word embeddings are sent into $L + 1$ ($0th - Lth$) transformer layers, where each transformer layer's output $h_i^l$ (layer $l$, position $i$) is the addition of previous layer's output $h_i^{l-1}$, this layer's multi-head self-attention (MHSA) layer output $A_i^l$, and this layer's feed-forward network layer (FFN) output $F_i^l$:

$$h_i^l = h_i^{l-1} + A_i^l + F_i^l \tag{1}$$

To compute the final distribution $Y$, the final layer's output at last position $h_T^L$ is multiplied with the unembedding matrix $E_u \in \mathbb{R}^{B \times d}$ and a softmax function over all $B$ tokens:

$$Y = softmax(E_u h_T^L) \tag{2}$$

As $h_T^L$ is the sum of the last position's layer outputs and previous studies (Olsson et al., 2022; Wang et al., 2023) find that attention layers play the largest roles for in-context learning, we focus on the last position $T$'s attention outputs. Each layer's MHSA output is computed by the weighted sum of different vectors:

$$A_T^l = \sum_{j=1}^{H} o_{j,T}^l \tag{3}$$

$$o_{j,T}^l = \sum_{p=1}^{T} \alpha_{j,T,p}^l \cdot O_j^l V_j^l h_p^{l-1} \tag{4}$$

$$\alpha_{j,T,p}^l = softmax(Q_j^l h_T^{l-1} \cdot K_j^l h_p^{l-1}) \tag{5}$$

where $o_{j,T}^l$ is the head output in head $j$, layer $l$. $\alpha_{j,T,p}^l$ is the attention score at position $p$, head $j$, layer $l$, computed by a softmax function over all positions' query-key inner products ($Q_j^l h_T^{l-1} \cdot K_j^l h_{pp}^{l-1}$, $pp$ from 1 to $T$). $V_j^l$ and $O_j^l$ are the value and output matrices in head $j$, layer $l$. Generally, $A_T^l$ can be regarded as the weighted sum of $H \times T$ value-output vectors over $H$ heads and $T$ positions, where $O_j^l V_j^l h_p^{l-1}$ is the value-output vector and $\alpha_{j,T,p}^l$ is its weight (attention score).

**Identifying important heads and important positions.** To explore the mechanism of in-context learning, Yu & Ananiadou (2024) identify the important heads for the final prediction token $b$ using causal interventions and log probability increase $S_j^l$ of each head output $o_{j,T}^l$:

$$S_j^l = log(p(b|o_{j,T}^l + h_T^{l-l})) - log(p(b|h_T^{l-1})) \tag{6}$$

If $S_j^l$ is large, it indicates that the head output $o_{j,T}^l$ contains important information about the final token $b$. Also, this importance score can be used to identify the important positions in this head by replacing $o_{j,T}^l$ with every position's weighted value-output vector $\alpha_{j,T,p}^l \cdot O_j^l V_j^l h_p^{l-1}$. They also design logit minus $M$ to evaluate the information storage of $o_{j,T}$ for two different tokens $b1$ and $b2$.

$$M = log(p(b1|o_{j,T})) - log(p(b2|o_{j,T})) \tag{7}$$

**Interpretability analysis: projecting vectors in unembedding space.** Geva et al. (2022) and Dar et al. (2022) find that many vectors are interpretable when projecting into the unembedding space $E_u$ by multiplying $E_u$ with the vectors. For instance, $EU_{j,T}^l$ is the projection of $o_{j,T}^l$.

$$EU_{j,T}^l = softmax(E_u o_{j,T}^l) \tag{8}$$

Yu & Ananiadou (2024) use this method to analyze the weighted value-output vectors in different positions and find that if $S_j^l$ is large for token $b$, $b$ usually ranks top in the projection $EU_{j,T}^l$.

## 3.2 MECHANISM EXPLORATION OF TEXTUAL QA

In this section, we explore the mechanism of TQA in Vicuna. We analyze 1,000 color-answering sentences of the form '[A] is [C]. Q: What is the color of [A]? A:', where [A] represents the animal and [C] represents the color. These sentences are derived from 1,000 images sampled from the COCO dataset (Lin et al., 2014). For VQA, the input consists of an image and the question 'Q: What is the color of [A]? A:'. The only difference between VQA and TQA is that, in the case of TQA, the image is translated into a textual context.

Inspired by previous studies (Olsson et al., 2022; Yu & Ananiadou, 2024), we propose the hypothesis shown in Figure 1(b) for TQA: In shallow layers, the color position extracts the animal information, while the last position encodes the question information. In deep layers' attention heads, the value-output matrices extract color information from the color position, and the query-key matrices compute the similarity between the last position's question features and the color position's animal features. When the question and the textual context refer to the same animal, the similarity score is high, leading to an increased probability of the color token in the final prediction's distribution.

We identify the most important heads and address four key questions related to our hypothesis: **a) Does the color position play the largest role in predicting the color token? b) Do the value-output matrices extract the color features from the color position? c) Does the color position extract the animal features from the textual context? d) Does the last position encode the animal features in the question?** To explore these questions, we design two comparison sentences: '[A1] is [C]. Q: What is the color of [A]? A:' and '[A] is [C]. Q: What is the color of [A1]? A:', where [A1] represents a different animal. We refer to the original sentence ('[A] is [C]. Q: What is the color of [A]? A:') as S0 and the comparison sentences as S1 and S2. The results are as below:

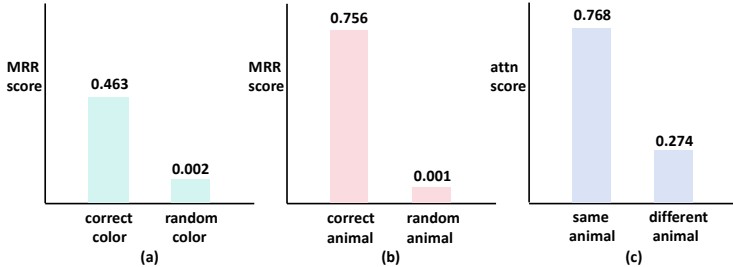

Figure 3: Analysis of color position's information storage in Vicuna TQA. (a) Color position value-output vector's information storage for correct color/random color. (b) Color position layer input vector's information storage for correct animal/random animal. (c) Color position's attention score when the question has the same/different animal with the textual context.

**Evidence a).** We calculate the proportion of the log probability increase at the color position relative to the total log probability increase across all positions. The proportion score is 99.82%, indicating that the color position plays the most significant role in predicting the final color token.

**Evidence b).** We compute the Mean Reciprocal Rank (MRR) of the color token's ranking when projecting the color position's weighted value-output vector (Eq.4) into the unembedding space (Eq.6), yielding an MRR score of 0.463 (equivalent to a ranking of 2.16). In comparison, a random color's MRR score is 0.002, as illustrated in Figure 3(a). The logit difference (Eq.7) between the correct color and a random color at the color position is 2.56. These results confirm that the value-output matrices effectively extract the color features from the color position.

**Evidence c).** Following Dar et al. (2022), we project the color position's layer input vector $h_p^{l-1}$ into the unembedding space and calculate the MRR score for the animal tokens [A] and [A1]. In S0, the MRR for [A] is 0.756, while for [A1], it is 0.001, as shown in Figure 3(b). The logit difference between [A] and [A1] at the color position is 0.32. In S1, the MRR for [A] is 0.002, the MRR for [A1] is 0.715, and the logit difference between [A1] and [A] is 1.70. These scores demonstrate that the color position's layer input vector in the most important heads encodes the animal features from its context.

**Evidence d).** We calculate the attention scores at the color position for S0, S1, and S2, as queried by the last position. The average attention scores are 0.768, 0.268, and 0.279 for S0, S1, and S2, respectively. When the question involves the same animal as the textual context, the attention score at the color position is high. However, when the animals differ, the attention score drops significantly, as shown in Figure 3(c). This drop in attention scores indicates that the last position encodes the question's animal features.

**Conclusion.** Based on the experimental results, we conclude: In shallow layers, the color position extracts the animal features from the textual context (evidence c), while the last position encodes the question features (evidence d). In deep layers' attention heads, the value-output matrices extract the color features from the color position (evidence b), and the query-key matrices compute the similarity score between the color position's animal features and the last position's question features (evidence d). When the question references the same animal as the textual context, the attention score is significantly high, resulting in the color position's weighted value-output vector containing substantial color information (evidence a), which is crucial for accurately predicting the color token.

## 3.3 MECHANISM EXPLORATION OF VISUAL QA

In this section, we aim to explore the mechanism of VQA in Llava. For VQA, we identify the most important heads and address the following questions: **a) What are the most important positions for predicting the correct color? b) Do the value-output matrices play a similar role as in TQA? c) Do the query-key matrices play a similar role as in TQA?** The results are as below:

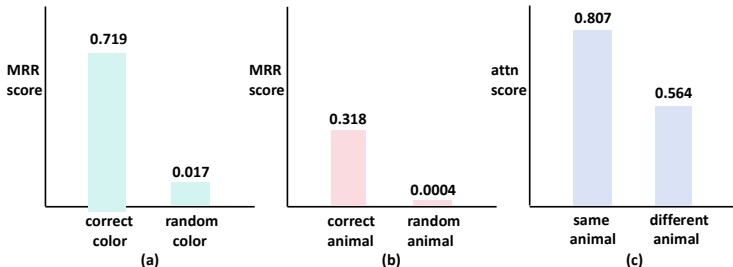

Figure 4: Analysis of top20 important positions' information storage in Llava VQA. (a) Top20 position value-output vectors' information storage for correct color/random color. (b) Top20 position layer input vectors' information storage for correct animal/random animal. (c) Top20 positions' sum attention score when the question has the same/different animal with the image.

**Evidence a).** We calculate the log probability increase for all positions and visualize these increases as heat maps overlaid on the corresponding images, as shown in Figure 2. After randomly sampling 200 sentences and analyzing the heat maps on a case-by-case basis, we observe that the positions with the largest log probability increases are those corresponding to image patches related to the animals. For example, when the question is "What is the color of the dog?" (as shown in Figure 2), the image patches related to the dog's head exhibit the largest log probability increase. This indicates that these positions contain crucial information for predicting the correct color, demonstrating strong interpretability. This observation inspired the design of an interpretability tool in Section 4, which helps explain why the model arrives at its final predictions. In contrast, the average attention score across all heads typically does not offer the same level of interpretability. Additional examples are provided in Appendix A.

**Evidence b).** After identifying the most important positions, we analyze whether the value-output matrices extract the color features from the top 20 important positions using a method similar to that used in TQA. When projecting the weighted value-output vector from the color position into the unembedding space, the MRR score for the correct color is 0.719 (equivalent to a ranking of 1.4) and the random color's MRR is 0.017, as shown in Figure 4(a). The logit difference between the correct color and a random color is 0.09. These results indicate that the value-output matrices effectively extract the color features from the top 20 important positions for the predicted color.

**Evidence c) and d).** We project the layer inputs of the top 20 important positions into the unembedding space and compute the MRR scores and logit differences between the correct animal and a

different animal. The correct animal's MRR score is 0.318, while the other animal's MRR score is 0.0004, as shown in Figure 4(b). The logit difference is 1.53, confirming that the important heads' layer inputs contain crucial information about the animals. Furthermore, when the animal in the question is replaced with another animal, the attention score at the top 20 positions drops significantly from 0.807 to 0.564 (see Figure 4c), indicating that the last position encodes information about the question.

**Similarity between VQA and TQA.** Our findings indicate that the mechanisms underlying Visual Question Answering (VQA) and Textual Question Answering (TQA) in deep layers are strikingly similar. In both cases, the layer inputs at key positions (the color position in TQA and the animal patch positions in VQA) contain essential information about the animal and color. The value-output matrices are responsible for extracting color information, while the query-key matrices compute the similarity of the animal information between these important positions and the last position. When the attention score is high, more of the color information from these positions is transferred to the last position, which, in turn, increases the likelihood of accurately predicting the color token.

**Evidence e). Difference between VQA and TQA: Information Contained in Visual Embeddings.** A key difference between Llava's VQA and Vicuna's TQA lies in the input embeddings at the 0th layer. In Vicuna, all input embeddings are word embeddings, whereas in Llava, the inputs are a mix of image embeddings and word embeddings (see Figure 1 a and c). In Vicuna, the color position contains color information, and the animal position contains animal information. When the color position's word embedding is projected into the embedding matrix $E$, the color token ranks first. A crucial question arises: Can this method be applied to analyze information stored in visual embeddings? Specifically, will the color and animal tokens rank highly when the visual embeddings are projected into the embedding matrix $E$? To investigate this, we projected the top 20 positions' visual embeddings into embedding space $E$ and computed the MRR scores for different tokens. The MRR for the correct color versus a random color is 0.455 versus 0.013, and for the correct animal versus a random animal, it is 0.076 versus 0.0003. We also calculated the MRR for the correct color and correct animal at a random position, which were 0.003 and 0.004, respectively. These results suggest that the top 20 positions already contain substantial information about the correct animal and color, whereas the random position does not.

**Conclusion.** Based on the experimental results, we propose the hypothesis about the mechanism of VQA illustrated in Figure 1(c). The visual embeddings generated by the projection $W$ and the CLIP visual encoder already contain information about the animal and the color (evidence e). This information is propagated through the positions' residual streams into the deep layers. In the deep layers' attention heads, the value-output matrices extract color information (evidence b), while the query-key matrices compute the similarity between the animal information and the question information at the last position (evidence c and d). When the similarity is high, the color information related to the animal in the question is more effectively transferred to the last position, thereby increasing the probability of correctly predicting the color token.

### 3.4 LLAVA'S VISUAL INSTRUCTION TUNING ENHANCES EXISTING ABILITIES OF VICUNA

In this section, we investigate how Llava acquires its VQA capabilities for color prediction. Building on our previous analysis, which highlighted the significant role of deep-layer attention heads in storing VQA abilities, we examine how the important heads evolve after visual instruction tuning. We compute the normalized importance scores for all heads and sort these scores for Vicuna TQA, Llava TQA, and Llava VQA. Figure 3 displays the importance of all 1,024 heads. In this visualization, the horizontal axis represents the layer number, while the vertical axis denotes the head number. The color intensity indicates the importance of each head, with darker colors signifying greater importance. Additionally, we list the top 10 heads for comparative purposes, where a label like 19_15 refers to the 15th head in the 19th layer.

When comparing Llava TQA with Vicuna TQA, we observe that 8 out of the top 10 heads are the same in both models. The remaining two heads also rank in the top 20 of the other model. In both models, the importance scores of the top10 heads are from 3.1% to 8.6%. In comparing Llava VQA with Llava TQA, we also find that 8 out of the top 10 heads are shared between the two. A significant difference is the sharp increase in the importance of head 19_6 (layer 19, head 6), which rises from 6.2% to 19.4%. This suggests that the importance of heads in Llava VQA

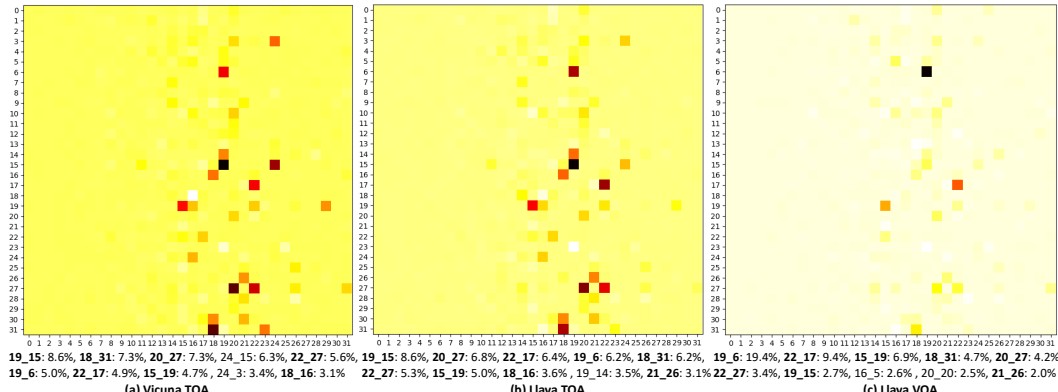

**19_15**: 8.6%, **18_31**: 7.3%, **20_27**: 7.3%, 24_15: 6.3%, **22_27**: 5.6%,   **19_15**: 8.6%, **20_27**: 6.8%, **22_17**: 6.4%, **19_6**: 6.2%, **18_31**: 6.2%,   **19_6**: 19.4%, **22_17**: 9.4%, **15_19**: 6.9%, **18_31**: 4.7%, **20_27**: 4.2%,
**19_6**: 5.0%, **22_17**: 4.9%, **15_19**: 4.7%, 24_3: 3.4%, **18_16**: 3.1%    **22_27**: 5.3%, **15_19**: 5.0%, **18_16**: 3.6%, 19_14: 3.5%, **21_26**: 3.1% **22_27**: 3.4%, **19_15**: 2.7%, 16_5: 2.6%, 20_20: 2.5%, **21_26**: 2.0%

         **(a) Vicuna TQA**                        **(b) Llava TQA**                      **(c) Llava VQA**

Figure 5: Top10 important heads in Vicuna TQA, Llava TQA and Llava VQA.

is more concentrated compared to Llava TQA. Based on these results, we conclude that: a) The important heads remain largely consistent between Llava TQA and Vicuna TQA. b) While the most crucial heads are generally similar between Llava VQA and Llava TQA, some heads, such as 19_6, become significantly more critical for VQA. c) Visual instruction tuning enhances the existing color-predicting ability of Vicuna's heads.

Overall, we explore the mechanism of TQA in Vicuna in Section 3.2 and that of VQA in Llava in Section 3.3. We find the mechanism of VQA and TQA is similar in the deep layers' attention heads. Furthermore, we analyze the projections of visual embeddings in the embedding matrix and find the visual embeddings already contain the information about the animals and the colors. Finally, we compared the most important heads in Vicuna TQA, Llava TQA and Llava VQA, and find that Llava enhances the existing heads' color predicting ability in Vicuna during visual instruction tuning.

## 4 INTERPRETABILITY TOOL FOR VISUAL QA

In this section, we present our interpretability tool for identifying the key image patches that influence the final predictions. This tool will be available for public use.

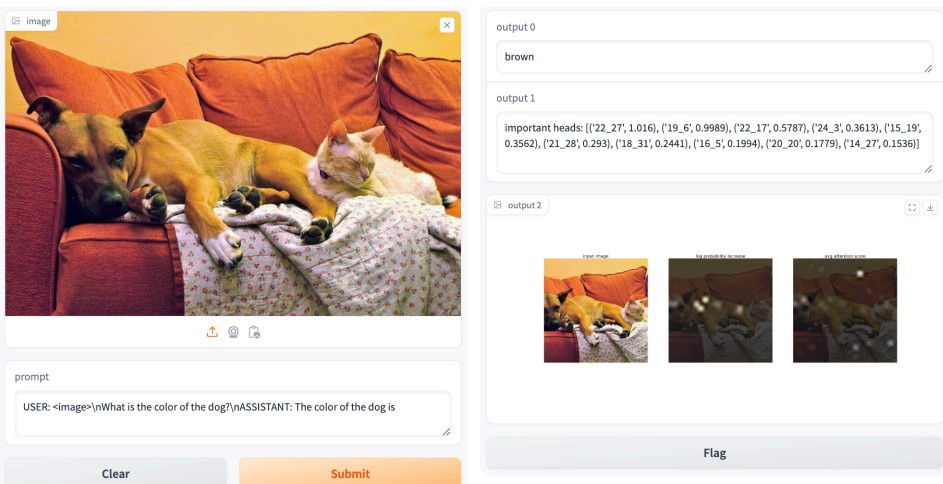

Figure 6: Interface of the interpretability tool. Left: image/question. Right: answer/visualization.

**Interface of the interpretability tool.** The interface is illustrated in Figure 6, developed using Gradio (Gradio, 2024). On the left side of the screen, users can upload an image and input a question. On the right side, the first box displays the prediction token, while the second box highlights the top 10 important heads related to the prediction. The third box shows the cropped image (the actual input

to Llava) along with the important image patches identified by log probability increase and average attention scores. Each image is divided into 24 x 24 image patches, with lighter areas indicating a larger score in log probability or attention. Although the visualization appears small within the interface, a button allows users to enlarge the images, resembling the first three images in Figure 2.

**Advantage 1: low computational cost.** The first advantage of our method is its low computational cost compared to causal explanations (Rohekar et al., 2024). Causal explanations typically require intervening on each image patch and calculating the impact on the final prediction, necessitating 24 x 24 + 1 model computations. In contrast, our method only requires a single model computation, with the internal vectors generated during the model's inference, resulting in minimal additional computation. With our approach, all computations can be completed within 2 seconds with one A100 GPU, offering a promising pathway for real-time explanations.

**Advantage 2: better interpretability.** Average attention score (Stan et al., 2024) across all attention heads is a widely used method for visual explanation. However, we have observed that this approach does not always provide reasonable explanations. For example, in Figure 6, when asked the question, 'What is the color of the dog?', the average attention score is higher on the pillow rather than on the dog itself. This suggests that the average attention score may fail to pinpoint the true reason behind the final prediction. In contrast, our method can accurately identify the important image patches related to the dog. The interpretability of these patches, identified by the log probability increase score, is grounded in the analysis from Section 3, offering a more reliable and robust understanding.

**Advantage 3: understanding visual hallucination.** Hallucination in vision-language models is a significant issue that has been extensively studied (Li et al., 2023; Zhou et al., 2023; Bai et al., 2024; Liu et al., 2024a). Understanding the precise cause of visual hallucination is crucial. For example, Figure 7 illustrates a hallucination case from Huang et al. (2024). When asked, 'What is the color of the left bottle?', Llava incorrectly answers 'Red'. The exact cause of the hallucination is unclear—whether the model misunderstood the word 'left' and provided the color of the right bottle, or if it simply returned the wrong color for the left bottle. Our method's interpretation clarifies that the model focuses on the bottom of the left bottle, revealing that the hallucination stems from the model failing to consider enough relevant image patches for the color, rather than from a misunderstanding of 'left' and 'right'. Furthermore, our interpretability method is versatile and can be applied to questions beyond color identification, as provided in Appendix A.

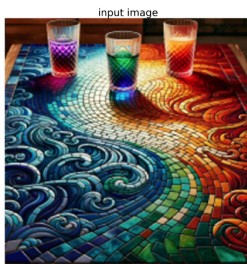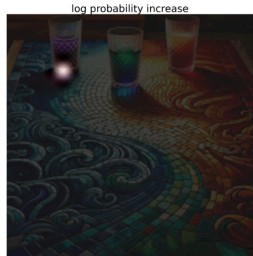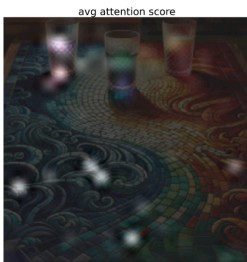

Figure 7: Understanding visual hallucination. Q: What is the color of the left bottle? A: **Red**

## 5 CONCLUSION

In this paper, we utilize mechanistic interpretability methods to investigate the mechanism of VQA in Llava. We find that the mechanism of VQA is similar to that of TQA. The visual embeddings encode the information of the animals and the colors, and the last position encodes the information of the question in shallow layers. In deep layers' attention heads, the value-output matrices extract the color information from the visual embeddings, and the query-key matrices compute the similarity between the last position's question features and the visual positions' animal features, controlling the probability of the final prediction. Moreover, we find that Llava enhances existing abilities of Vicuna during visual instruction tuning. Based on this analysis, we design an interpretability tool for locating the important image patches related to the final prediction, which has low computational cost, better interpretability and can be utilized for understanding visual hallucination. Overall, our method and analysis is helpful for understanding the mechanism of VQA.

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

## A    APPENDIX A: EXAMPLE IMAGES' INTERPRETABILITY

In this section, we provide more examples to verify the usage of our interpretability tool. Our method is not only suitable for identifying the important image patches about color questions, but also for other questions. Compared with average attention, our method usually expresses much better interpretability. The questions are listed in the titles of the following images, where the answers are marked as bold.

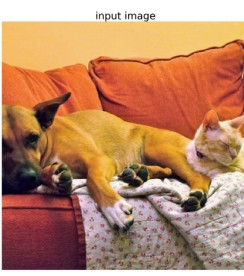 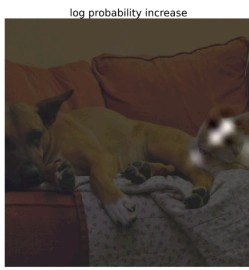 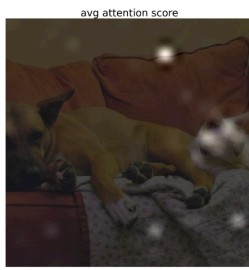

Figure 8: Q: What is the color of the cat? A: The color of the cat is **white**

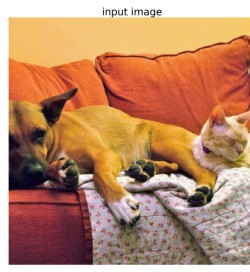 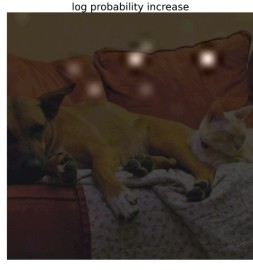 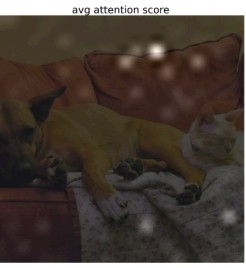

Figure 9: Q: What is the color of the pillow? A: The color of the pillow is **orange**

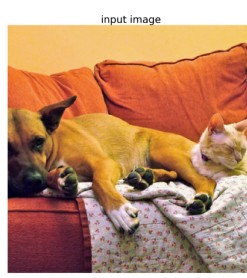 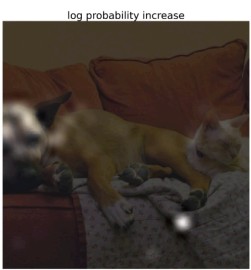 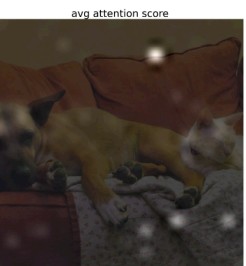

Figure 10: Q: What is the left animal? A: The left animal is a **dog**

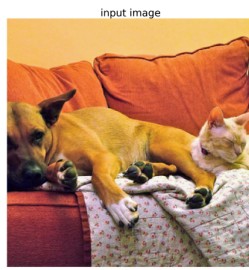 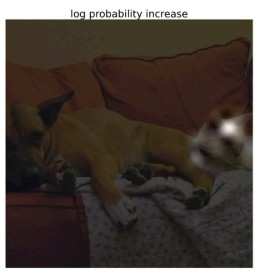 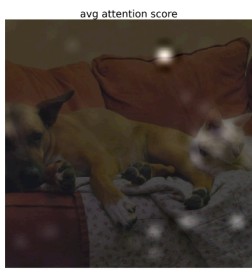

Figure 11: Q: What is the right animal? A: The right animal is a **cat**

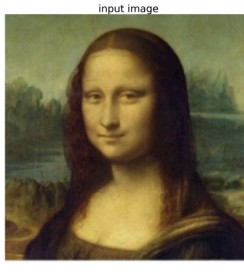 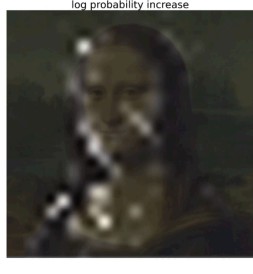 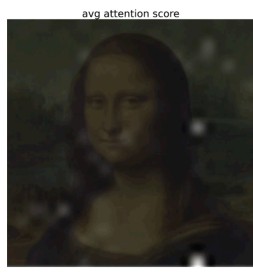

Figure 12: Q: What is in the painting? A: The painting features a **woman**

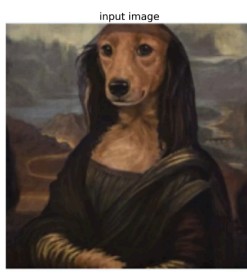 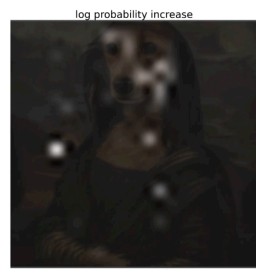 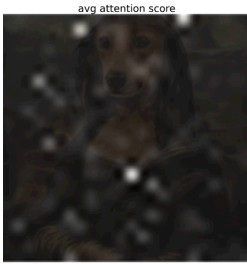

Figure 13: Q: What is in the painting? A: The painting features a **dog**

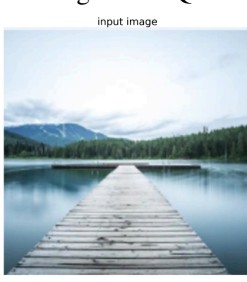 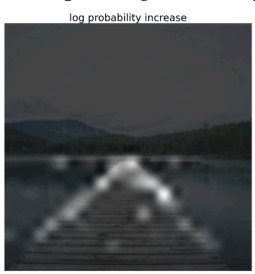 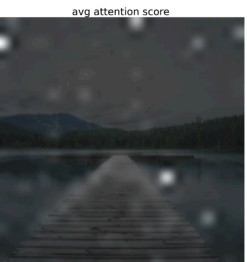

Figure 14: Q: What is in the picture? A: The picture features a **pier**

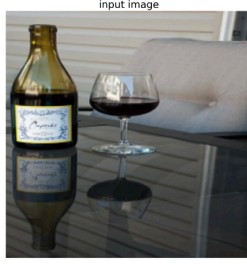 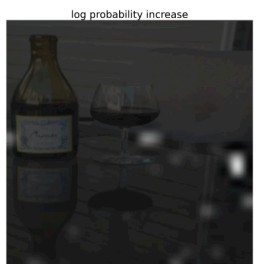 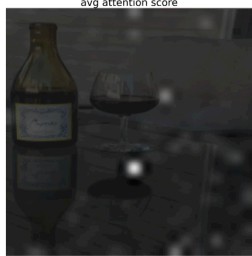

Figure 15: Q: What is the table made of? A: The table is made of **glass**