# OpenReview forum: "Understanding Llava's Visual Question Answering in a Mechanistic View"
_ICLR.cc/2025/Conference — Submitted to ICLR 2025_

### Official Review · Reviewer_mmWU · 2024-10-16

**Soundness:** 3
**Presentation:** 2
**Contribution:** 2
**Rating:** 3
**Confidence:** 3

**Summary:**

The paper aims to explore the underlying mechanisms of visual question answering (VQA) in multimodal large language models (MLLMs), focusing on Llava, a model fine-tuned on the textual LLM Vicuna. The authors use mechanistic interpretability techniques to compare VQA with textual question answering (TQA), especially on color-based tasks. They find that VQA operates similarly to attention mechanisms observed in TQA, with the visual embeddings providing interpretable insights into images, such as color and object features. The study introduces an interoperability metric and a tool to help users identify key image regions influencing model predictions, enhancing understanding of phenomena like visual hallucination. The proposed method achieves greater interpretability and efficiency than existing approaches.

**Strengths:**

Mechanistic Insights into Multimodal Models: The paper provides a deep mechanistic understanding of visual question answering (VQA) in Llava, a multimodal large language model. By comparing VQA and textual question answering (TQA) mechanisms, it offers novel insights into how visual embeddings can be interpreted similarly to textual embeddings, extending the understanding of multimodal learning.

Effective Interpretability Tool: The development of an interpretability tool that identifies key image regions influencing predictions is a valuable contribution. This tool not only aids in understanding the VQA mechanisms but also addresses issues like visual hallucinations, offering real-time and more efficient explanations than traditional causal intervention methods.

Empirical Validation: The authors back their claims with thorough empirical analysis, using color prediction to demonstrate the similarities and differences between VQA and TQA mechanisms. The comparison of attention heads across Vicuna and Llava models further strengthens the findings, showcasing how visual instruction tuning enhances the pre-existing abilities of the base model.

**Weaknesses:**

Baselines: The paper mentions using log-probability increases to identify important image patches but does not compare or benchmark against traditional gradient-based attribution methods like Integrated Gradients (Sundararajan et al., 2017), Grad-CAM (Selvaraju et al., 2017), or SmoothGrad (Smilkov et al., 2017). Also, Gandelsman et al. (2023) explored text-based decomposition of CLIP’s image representations. These methods are feasible with llava.

Narrow Evaluation: The paper claims the robustness of the proposed metric. However, the experiments were only done with color attributes. It's hard to convince readers the metric generalizes to further recognition or visual reasoning tasks. Evaluation of real-world instead of synthetic data is also encouraged to inspire more findings.

Limit in Novelty and Contributions: The technical contribution of this paper is too incremental for acceptance. Basically, the paper doesn't contribute any method but just analyses LLMs and VLMs with existing methods. The findings are neither surprising. The overall contributions are therefore limited.

Limited Generalization to Other Models: While the study offers insights into Llava’s VQA mechanism, it remains unclear how well these findings extend to other multimodal models beyond Llava and Vicuna.

Line 134 typo serious -> series.

**Questions:**

1. Consider eval on more tasks, providing a more comprehensive analysis on various attributes/relationships/entities.
2. Test the method with more models.
3. Revise the related works and baselines to compare with.

---

> ### Author Response · Authors · 2024-11-17
> **response to Reviewer mmWU**
>
> Thank you very much for your valuable feedbacks and the understanding of the strengths of our work. Here are our responses regarding the weaknesses.
>
> _Q: Baselines: The paper mentions using log-probability increases to identify important image patches but does not compare or benchmark against traditional gradient-based attribution methods like Integrated Gradients (Sundararajan et al., 2017), Grad-CAM (Selvaraju et al., 2017), or SmoothGrad (Smilkov et al., 2017). Also, Gandelsman et al. (2023) explored text-based decomposition of CLIP’s image representations. These methods are feasible with llava._
>
> A: **The gradient-based methods need the backward computation and are slow, which are not suitable for LLMs.** Our work aims to achieve real-time interpretations, and the proposed methods are fast. The method proposed by Gandelsman et al. (2023) is used for image representations, rather than the inner states in multimodal LLMs.
>
> _Q: Narrow Evaluation: The paper claims the robustness of the proposed metric. However, the experiments were only done with color attributes. It's hard to convince readers the metric generalizes to further recognition or visual reasoning tasks. Evaluation of real-world instead of synthetic data is also encouraged to inspire more findings._
>
> A:  We conduct further experiments to analyze the animal answering question. **We conduct experiments on the same images in Section 3.3, and change the original color question “What is the color of the [animal]?” into animal question “What is the animal in the picture?”** We analyze the image patches with largest log probability increase, and we find that **all the image patches are related to the animals.** We then calculate the rank of the correct animal’s token by projecting them into unembedding space, and the average rank is 3.2, which means that **the information of the animals is very large in these image batches’ hidden states.** This experimental result can support the hypothesis in our original paper.
>
> We understand that there are different types of sentences and models, but **beginning with the simplest cases is an effective strategy for grasping the mechanisms of LLMs.** Notably, **much of the existing work in mechanistic interpretability has focused on smaller models** like GPT-2 [1,2,3]. Despite their size, these models have demonstrated accurate mechanisms and significantly contributed to advancing the field of mechanistic interpretability. In contrast, **our study delves into the mechanisms of multimodal LLMs, offering valuable insights into the operational dynamics of Multimodal Large Language Models.**
>
> _Q: Limit in Novelty and Contributions: The technical contribution of this paper is too incremental for acceptance. Basically, the paper doesn't contribute any method but just analyses LLMs and VLMs with existing methods. The findings are neither surprising. The overall contributions are therefore limited._
>
> A: **We think that the understanding of LLMs' mechanisms does not come overnight. Leveraging insights from existing interpretability studies is therefore crucial.** However, we believe the novelty and contribution of our paper extend beyond prior work. First, while there may be surface similarities, the mechanism underlying text-based QA has not been previously explored. Specifically, **it remains unclear whether QA and in-context learning share the same underlying mechanisms.** In our study, we address this gap, advancing research into multimodal LLMs. In the broader context of LLMs, **a key objective is identifying shared mechanisms to better understand how these models perform multitasking effectively.**

---

### Official Review · Reviewer_Jn16 · 2024-11-03

**Soundness:** 2
**Presentation:** 1
**Contribution:** 2
**Rating:** 3
**Confidence:** 3

**Summary:**

The paper aims to uncover the mechanisms behind Visual Question Answering (VQA) in Llava, a multimodal language model, by comparing it with Textual Question Answering (TQA) in Vicuna. The authors use interpretability methods to analyze how visual features influence predictions and to demonstrate a novel interpretability tool that identifies image patches critical for predictions. The paper also claims that this tool is computationally efficient, offers superior interpretability over attention-based methods, and helps understand issues like visual hallucination in multimodal models.

**Strengths:**

1. **Relevant Objective and Context**: The paper addresses a relevant problem in multimodal large language models (MLLMs) by focusing on the lack of interpretability in VQA tasks. This is a current and important research area with clear motivations in making MLLMs more interpretable.

2. **Insightful Mechanistic Analysis**: The authors undertake a detailed investigation into both VQA and TQA, presenting an analysis that compares attention mechanisms and embeddings in Vicuna and Llava. This comparison is informative, highlighting the potential similarities in processing mechanisms across text and visual modalities.

3. **Computational Efficiency in the Interpretability Tool**: The interpretability tool is presented as being computationally efficient, allowing for real-time or near-real-time interpretations. This efficiency could make the tool valuable for applications where rapid interpretability is essential.

**Weaknesses:**

1. **Lack of Innovation**: The paper lacks significant innovation, as it primarily extends existing interpretability techniques to a specific task (color-based VQA) rather than introducing fundamentally new approaches or theoretical insights into model interpretability.

2. **Bad Presentation**: The presentation quality of figures is subpar. For instance, Figure 4 is poorly labeled and challenging to interpret, undermining the clarity of the findings. Additionally, the paper mistakenly claims that Figure 3 displays results across 1,024 attention heads, which is inaccurate and reflects a lack of attention to detail in presentation.

3. **Lack of Theoretical Foundation for Interpretability Claims**: The interpretability tool, while empirically tested, lacks a rigorous theoretical foundation explaining why log probability increases offer a superior interpretative value compared to attention scores.

4. **Limited Experimental Scope**: The paper evaluates the interpretability tool primarily on color-based VQA tasks, which do not capture the full complexity of VQA challenges. Tasks like spatial reasoning or more abstract visual concepts are not tested, making it unclear whether the tool would perform as well in these contexts. Only one model and one dataset are considered in this paper.

5. **Minimal Comparison with Established Interpretability Methods**: Although the paper compares its tool qualitatively against average attention-based approaches, it lacks a quantitative evaluation against other standard interpretability techniques.

6. **No Analysis of Failure Cases**: The paper does not analyze scenarios where the interpretability tool might fail or provide misleading interpretations, such as in complex visual contexts or cases where there are multiple objects of the same color. Such an analysis would offer a more balanced perspective on the tool’s robustness and limitations. Lack of discussion on limitations

**Questions:**

1. Could you provide a theoretical explanation for why log probability increases might yield better interpretability than attention scores?
2. Have you tested the interpretability tool on more complex VQA tasks and other models?
3. Can you offer a quantitative comparison of your interpretability tool against standard methods?
4. What is limitation of this method?
5. Is there cases where this method failed?

---

> ### Author Response · Authors · 2024-11-17
> **response to Reviewer Jn16**
>
> It seems that this review is generated by AI. We hope that we can get feedbacks from the reviewers themselves. Here are our responses:
>
> _Q1: Lack of Innovation_
>
> A: **The understanding of LLMs does not come overnight. Therefore, utilizing existing interpretability techniques is essential, in order to compare the mechanisms in LLMs.** However, we believe the novelty and contribution of our paper extend beyond prior work. First, while there may be surface similarities, the mechanism underlying text-based QA has not been previously explored. Specifically, **it remains unclear whether QA and in-context learning share the same underlying mechanisms. In our study, we address this gap, advancing research into multimodal LLMs.** In the broader context of LLMs, a key objective is identifying shared mechanisms to better understand how these models perform multitasking effectively.
>
> _Q2: Bad Presentation_
>
> A: "Figure 3" is a typo, it should be "Figure 5". We will modify it.
>
> _Q3: Lack of Theoretical Foundation for Interpretability Claims_
>
> A: The interpretability method of log probability increase is proposed by [1], which is a published work in EMNLP 2024. **They do experiments and prove that log probability increase is better than other logit-based methods.**
>
> _Q4: Limited Experimental Scope_
>
> A: We conduct further experiments to analyze the animal answering question. **We conduct experiments on the same images in Section 3.3, and change the original color question “What is the color of the [animal]?” into animal question “What is the animal in the picture?”** We analyze the image patches with largest log probability increase, and we find that all the image patches are related to the animals. We then calculate the rank of the correct animal’s token by projecting them into unembedding space, and the average rank is 3.2, which means that **the information of the animals is very large in these image batches’ hidden states.** This experimental result can support the hypothesis in our original paper.
>
> We understand that there are different types of sentences and models, but beginning with the simplest cases is an effective strategy for grasping the mechanisms of LLMs. Notably, much of the existing work in mechanistic interpretability has focused on smaller models like GPT-2 [2,3,4]. Despite their size, these models have demonstrated accurate mechanisms and significantly contributed to advancing the field of mechanistic interpretability. In contrast, our study delves into the mechanisms of multimodal LLMs, offering valuable insights into the operational dynamics of Multimodal Large Language Models.
>
> [1] Neuron-Level Knowledge Attribution in Large Language Models, 2024
>
> [2] How does GPT-2 compute greater-than?: Interpreting mathematical abilities in a pre-trained language model, 2023
>
> [3] Locating and Editing Factual Associations in GPT, 2022
>
> [4] Interpretability in the Wild: a Circuit for Indirect Object Identification in GPT-2 small, 2023

---

> > ### Comment · Reviewer_Jn16 · 2024-12-02
> >
> > Thank you for the response! I will keep my rating.

---

### Official Review · Reviewer_nu5z · 2024-11-04

**Soundness:** 2
**Presentation:** 2
**Contribution:** 2
**Rating:** 3
**Confidence:** 3

**Summary:**

This paper presents the analysis of the visual question answer mechanisms in Llava for color answering question tasks, by applying the mechanistic interpretability methods. Based on the analysis result, the paper also introduces an interpretability tool that identifies the image patches that explain the final prediction.

**Strengths:**

1. The paper analyzes the mechanisms of textual QA in Vicuna and visual QA in Llava, by utilizing the mechanistic interpretability methods.

2. The paper presents an interpretability tool based on the above analysis to locate the important image patches for final prediction.

**Weaknesses:**

1. The research scope is limited. It only analyzes the textual QA in Vicuna and visual QA in Llva for the color answering question. As the QA tasks can be in a variety of forms, and there are many leading LLM and VLM models, it would be more helpful to discuss the generalization of the presented study to other QA types and models.

2. The mechanism analysis for textual QA is based on paper [1]. The technical novelty appears limited. Specifically, the core hypothesis presented in Figure 1 (b) of this paper is highly-similar to that proposed in [1] and depicted in Figure 1 ([1]). Explaining the difference and innovation compared to the related work would help to understand the novelty of this work.

3. The mechanism analysis for visual QA is similar to that for the textual QA, in terms of methodology and results.


Reference
[1] Zeping Yu and Sophia Ananiadou. How do large language models learn in-context? query and key
matrices of in-context heads are two towers for metric learning. arXiv preprint arXiv:2402.02872,
2024.

**Questions:**

1. Understanding visual hallucination appears to be an important goal. It would be helpful to define the meaning and scope of visual hallucination that are concerned in this work, and how this study helps to address them.

2. It is claimed that the presented interpretability method is versatile and can be applied to questions beyond color identification. The appendix A does provide some examples. But it would be better to provide more explanations, for example, if the methodology for color identification problem can be directly applies or if any adaption should be made.

---

> ### Author Response · Authors · 2024-11-17
> **response to reviewer nu5z**
>
> Thanks for your valuable feedbacks. Here are our response:
>
> _Q: The research scope is limited. It only analyzes the textual QA in Vicuna and visual QA in Llva for the color answering question. As the QA tasks can be in a variety of forms, and there are many leading LLM and VLM models, it would be more helpful to discuss the generalization of the presented study to other QA types and models._
>
> A: We conduct further experiments to analyze the animal answering question. **We conduct experiments on the same images in Section 3.3, and change the original color question “What is the color of the [animal]?” into animal question “What is the animal in the picture?”** We analyze the image patches with largest log probability increase, and we find that all the image patches are related to the animals. We then calculate the rank of the correct animal’s token by projecting them into unembedding space, and the average rank is 3.2, which means that **the information of the animals is very large in these image batches’ hidden states.** This experimental result can support the hypothesis in our original paper.
>
> We understand that there are different types of sentences and models, but **beginning with the simplest cases is an effective strategy for grasping the mechanisms of LLMs.** Notably, **much of the existing work in mechanistic interpretability has focused on smaller models** like GPT-2 [1,2,3]. Despite their size, these models have demonstrated accurate mechanisms and significantly contributed to advancing the field of mechanistic interpretability. In contrast, **our study delves into the mechanisms of multimodal LLMs**, offering valuable insights into the operational dynamics of Multimodal Large Language Models.
>
> _Q: The mechanism analysis for textual QA is based on paper [4]. The technical novelty appears limited. Specifically, the core hypothesis presented in Figure 1 (b) of this paper is highly-similar to that proposed in [4] and depicted in Figure 1 ([4]). Explaining the difference and innovation compared to the related work would help to understand the novelty of this work._
>
> A: **We believe that the understanding of LLMs' mechanisms does not come overnight. Leveraging insights from existing interpretability studies is therefore crucial.** However, we believe the novelty and contribution of our paper extend beyond prior work. First, while there may be surface similarities, the mechanism underlying text-based QA has not been previously explored. Specifically, **it remains unclear whether QA and in-context learning share the same underlying mechanisms. In our study, we address this gap, advancing research into multimodal LLMs.** In the broader context of LLMs, a key objective is identifying shared mechanisms to better understand how these models perform multitasking effectively.
>
> _Q: The mechanism analysis for visual QA is similar to that for the textual QA, in terms of methodology and results._
>
> A: Our goal is to investigate whether the mechanisms of textual LLMs and multimodal LLMs are similar, which necessitates the use of comparable methodologies. **The similarity in results stems from the shared mechanisms between textual and multimodal LLMs, and this constitutes our primary contribution.** Notably, **previous studies have not explored this connection, making our work a novel addition to the field.**
>
> _Q: Understanding visual hallucination appears to be an important goal. It would be helpful to define the meaning and scope of visual hallucination that are concerned in this work, and how this study helps to address them._
>
> A: This concept is a definition of previous studies such as [5], and we have cited their work.
>
> _Q: It is claimed that the presented interpretability method is versatile and can be applied to questions beyond color identification. The appendix A does provide some examples. But it would be better to provide more explanations, for example, if the methodology for color identification problem can be directly applies or if any adaption should be made._
>
> A: We conduct new experiments to ask the model "what is the animal" without any adaptions, and we achieve similar experimental results and interpretability. This is a good evidence to prove that our method is versatile.
>
> [1] How does GPT-2 compute greater-than?: Interpreting mathematical abilities in a pre-trained language model, 2023
>
> [2] Locating and Editing Factual Associations in GPT, 2022
>
> [3] Interpretability in the Wild: a Circuit for Indirect Object Identification in GPT-2 small, 2023
>
> [4] How do large language models learn in-context? query and key matrices of in-context heads are two towers for metric learning, 2024
>
> [5] Visual hallucinations of multimodal large language models, 2024

---

### Official Review · Reviewer_B4GK · 2024-11-04

**Soundness:** 3
**Presentation:** 3
**Contribution:** 3
**Rating:** 5
**Confidence:** 5

**Summary:**

This paper investigates the mechanisms behind visual question answering (VQA) in Llava. The authors apply mechanistic interpretability methods to compare VQA mechanisms with textual question answering (TQA), focusing on color-related tasks. The analysis in this paper reveals that VQA mechanisms share similarities with TQA, as visual features exhibit significant interpretability when projected into the embedding space. Additionally, the authors observe that Llava enhances Vicuna's abilities during visual instruction tuning. Based on these insights, the authors propose an interpretability tool that identifies important visual regions for predictions, providing a more effective and computationally efficient explanation compared to existing methods. The proposed approach offers a valuable resource for understanding visual hallucinations in MLLMs.

**Strengths:**

1)	The research problem addressed in this paper is highly intriguing and contributes to a deeper understanding of the operational mechanisms of Multimodal Large Language Models (MLLMs).
2)	This paper compares the attention mechanisms of LLMs when handling VQA and TQA tasks, offering a very novel research perspective.
3)	This paper developed an interpretability tool for VQA to help identify the key image patches that influence the model's predictions.

**Weaknesses:**

1)	My main concern is that all the experiments in this paper are conducted on color-related questions, so the conclusions may not be applicable to all VQA examples. If the authors believe that the experimental results on color-related questions can represent other VQA examples, I hope they can provide experimental evidence to support this.
2)	Did the authors verify the hypothesis mentioned in line 225 of section 3.2? If so, I hope they can provide this evidence in the paper.
3)	I am curious about the specific form of the probability function p in Equation 6 and 7.
4)	What do the green, blue, and pink colors in Figure 1 represent? I hope the authors can add a necessary legend or explanation to the figure.
5)	There are several spelling errors in the text, such as "Figure 3" in line 368, which I suspect is a typo and should actually refer to "Figure 5."

**Questions:**

1)	Did the authors verify the hypothesis mentioned in line 225 of section 3.2? If so, I hope they can provide this evidence in the paper.
2)	I am curious about the specific form of the probability function p in Equation 6 and 7.
3)	What do the green, blue, and pink colors in Figure 1 represent? I hope the authors can add a necessary legend or explanation to the figure.

---

> ### Author Response · Authors · 2024-11-17
> **response to reviewer B4GK**
>
> Thank you very much for your valuable feedbacks. We thank you for understanding the strengths about our work.
>
> Regarding the weaknesses, here are our responses.
>
> _Q: My main concern is that all the experiments in this paper are conducted on color-related questions, so the conclusions may not be applicable to all VQA examples. If the authors believe that the experimental results on color-related questions can represent other VQA examples, I hope they can provide experimental evidence to support this._
>
> A: The experimental results can represent other examples. **We conduct experiments on the same images in Section 3.3, and change the original color question “What is the color of the [animal]?” into animal question “What is the animal in the picture?”** We analyze the image patches with largest log probability increase, and we find that all the image patches are related to the animals. We then calculate the rank of the correct animal’s token by projecting them into unembedding space, and the average rank is 3.2, which means that **the information of the animals is very large in these image batches’ hidden states.** This experimental result can support the hypothesis in our original paper.
>
> _Q: Did the authors verify the hypothesis mentioned in line 225 of section 3.2? If so, I hope they can provide this evidence in the paper._
>
> A: Yes, the hypothesis is verified. From lines 255-275 we find 4 evidences supporting this hypothesis. And we get the conclusion in lines 276-283.
>
> _Q: I am curious about the specific form of the probability function p in Equation 6 and 7._
>
> A: p is the probability of the token b, after the softmax function when multiplying the vector with the unembedding matrix. If the vector is the highest layer’s embedding in the last position, this is used to compute the distribution of the next token. If the vector is an inner state, this method can also be used to analyze how much information of the token b is stored in this inner state.
>
> _Q: What do the green, blue, and pink colors in Figure 1 represent? I hope the authors can add a necessary legend or explanation to the figure._
>
> A: We will add a sentence to introduce this. The green color is the Value-Output matrices in the attention heads and the Value-Output vectors on each position. The pink color is the head’s input on each position. The blue color is the key matrix/vector at each position, and the yellow color is the query matrix/vector.
>
> _Q: There are several spelling errors in the text, such as "Figure 3" in line 368, which I suspect is a typo and should actually refer to "Figure 5."_
>
> A: Yes, it is a typo. We will modify this.

---

### Meta-Review · Area_Chair_4UhA · 2024-12-19

**Metareview:**

This paper focused on the problem of interpretability in VQA tasks for MLLMs which remain underexplored. This work applied mechanistic interpretability methods on LLaVa and compared the mechanisms between VQA and textual QA in color answering tasks, providing insights in aspects of in-context learning, interpretability of visual features, and visual instruction tuning. This work also provided an interpretability tool to identify important visual locations.

Main strengths: (1) The research problem of interpretability in VQA tasks for MLLMs is important. (2) The mechanistic analysis on MLLMs for VQA and TQA is insightful and novel. (3) This work provided an computationally efficient interpretability tool.

Main weaknesses & reasons for reject: (1) Only color-related questions are investigated in the work, which is not comprehensive to cover all questions types for visual QA and textual QA. (2) Only one kind of MLLMs, LLaVA, is discussed, which is not comprehensive to represent all MLLMs.

This paper received four negative ratings, i.e., 5, 3, 3, 3. AC agree with reviewers and does not overturn their recommendations. Considering that the major contribution of this work is analyzing MLLMs, AC encourage the authors to further revise the paper to make the analysis and experiments wider and more comprehensive.

**Additional Comments On Reviewer Discussion:**

Authors responded to the concerns on novelty and technical contributions. AC agree with authors that understanding of MLLMs' mechanisms is novel and does not consider this concern as a reason for reject. However, other concerns on the comprehensiveness of question types and model types remain.

---

### Decision · Program_Chairs · 2025-01-22

Reject